# Impact of Fatty Acid-Binding Proteins in α-Synuclein-Induced Mitochondrial Injury in Synucleinopathy

**DOI:** 10.3390/biomedicines9050560

**Published:** 2021-05-17

**Authors:** An Cheng, Wenbin Jia, Ichiro Kawahata, Kohji Fukunaga

**Affiliations:** 1Departments of Pharmacology, Graduate School of Pharmaceutical Science, Tohoku University, Sendai 980-8578, Japan; cheng.an.q6@dc.tohoku.ac.jp (A.C.); jia.wenbin.t6@dc.tohoku.ac.jp (W.J.); kawahata@tohoku.ac.jp (I.K.); 2Department of CNS Drug Innovation, Graduate School of Pharmaceutical Science, Tohoku University, Sendai 980-8578, Japan

**Keywords:** α-synuclein, fatty acid-binding proteins, neurodegenerative disorders, mitochondria

## Abstract

Synucleinopathies are diverse diseases with motor and cognitive dysfunction due to progressive neuronal loss or demyelination, due to oligodendrocyte loss in the brain. While the etiology of neurodegenerative disorders (NDDs) is likely multifactorial, mitochondrial injury is one of the most vital factors in neuronal loss and oligodendrocyte dysfunction, especially in Parkinson’s disease, dementia with Lewy body, multiple system atrophy, and Krabbe disease. In recent years, the abnormal accumulation of highly neurotoxic α-synuclein in the mitochondrial membrane, which leads to mitochondrial dysfunction, was well studied. Furthermore, fatty acid-binding proteins (FABPs), which are members of a superfamily and are essential in fatty acid trafficking, were reported to trigger α-synuclein oligomerization in neurons and glial cells and to target the mitochondrial outer membrane, thereby causing mitochondrial loss. Here, we provide an updated overview of recent findings on FABP and α-synuclein interactions and mitochondrial injury in NDDs.

## 1. Introduction

Neurodegenerative disorders (NDDs) are characterized by the progressive deterioration of neurons, followed by degeneration of the structure and function of axons, dendrites, and synapses, and finally leading to neuronal death [1] or dysfunction of myelination [2]. Several diseases are under the umbrella of NDDs, most of which are Alzheimer’s disease (AD) and Parkinson’s disease (PD), caused by neuronal death in the brain [3,4], and other oligodendroglia-related disorders, such as multiple system atrophy (MSA) and Krabbe disease (KD). Mitochondria are key organelles for normal cellular functions and are involved in energy production, calcium homeostasis, as well as cell survival and mitochondrial injury. They were proposed to be the central pathogenesis of PD [5], MSA [6], KD [7], and some other NDDs [8,9]. Recently, the protein α-synuclein is in the spotlight of the mechanism of mitochondrial injury in NDDs. Its abnormal accumulation and aggregation in mitochondria [10] impairs complex I and causes oxidative stress [11] in human dopaminergic neuronal cultures and the PD brain [12]. In contrast, fatty acid-binding proteins (FABPs) are the new pathological factors involved in α-synuclein aggregation and migration in PD [13,14,15]. Importantly, some FABPs, such as FABP 1 [16], FABP 4 [17], and FABP 5 [18] are also localized in the mitochondria and are related to mitochondrial dysfunction.

In this review, we summarize the current data on the pathological functions of α-synuclein and FABPs in NDDs and focus on the underlying relationship between their interactions and ability in mitochondrial injury. We further discuss the role of FABP inhibition in cell protection and provide some novel and critical perspectives on NDD-therapy-targeting mitochondria.

## 2. α-Synuclein

### 2.1. α-Synuclein Structure and Mutations

α-Synuclein is a low-molecular weight protein that is composed of 140 amino acids and is divided into three distinct regions—a positively charged N-terminal region (1–60), a central hydrophobic region (61–95), which is known as the non-amyloid-beta component (NAC) and is involved in α-synuclein aggregation and a highly acidic C-terminal domain (96–140) [19,20]. α-Synuclein is an intrinsically disordered protein abundantly expressed in the central nervous system (CNS) [21] and was first purified in vitro in soluble and monomeric forms [22]. Usually, disordered proteins possess primary sequences enriched in charged residues and prolines [23] that avoid misfolding and aggregation. In the case of α-synuclein, the NAC domain is partially protected by the positive and negative charges of the N- and C-terminal regions. However, mutations change their environmental conditions and disturb the native compactness of α-synuclein, thereby triggering aggregation. To date, six point-mutations (A53T, A30P, A53E, E46K, G51D, and H50Q) of α-synuclein were reported to be associated with synucleinopathies (Figure 1) [24,25,26,27,28,29]. Their rates of aggregation and oligomerization were studied widely in vitro [30,31,32,33]. Among them, α-synuclein in E46K, H50Q, and A53T mutations are more prone to aggregation, while α-synuclein in the A30P mutation appeared to be more prone to oligomerization [34]. All mutations identified to date are located in the N-terminal domain but do not promote changes in the major structure to the α-synuclein monomer. Only the A30P mutation suggested a reduced propensity to adopt alpha-helical content [35]. The membrane-binding properties of mutation types are likely affected by their localization in the N-terminal domain. For example, A53T and H50Q suggest a higher propensity to bind with membranes [36]; however, A30P, A53E, and G51D suggest a lower propensity to bind with membranes [37,38,39]. Additionally, these mutations alter long-range electrostatic interactions between the N- and C-termini, thereby, affecting the stability of the native α-synuclein [40]. Therefore, aberrant internal contacts of mutated α-synuclein might cause a partially folded intermediate that increases the propensity for self-assembly [41].

### 2.2. α-Synuclein Oligomers and Neuronal Toxicity

The definition of α-synuclein oligomers is rather ambiguous, owing to their heterogeneity, instability, transient existence, and variability. Oligomers are regarded as intermediates of the aggregation process but suggest higher toxicity than the other forms [42,43,44] and cause a wide range of damaging effects, such as inhibition of exocytosis and dopamine release in PC12 cells [45], complex I-dependent, Ca^2+^-induced mitochondrial dysfunction in an in vitro system comprising isolated mitochondria [46], and altering synaptic transmission in primary hippocampal and dissociated nigral neurons [47]. Additionally, α-synuclein oligomers can permeabilize lipid bilayers [48,49] and alert plasma and intracellular membrane structures [50], by rearranging the lipid microenvironment and accelerating the flux of hydrophilic molecules in the surroundings [49,51]. On the other hand, α-synuclein oligomers also induce pore formation and act as pathological membrane channels, while tubular amyloid or ring-shaped oligomers are integrated into the membrane [48,52]. Although α-synuclein oligomers seem like they connect with plasma membrane in neurons, the structural characterization of the putative pores remains unclear. Furthermore, it was reported that α-synuclein oligomers inhibit tubulin polymerization, affect the cytoskeleton of cells, and damage the integrity of the neurite network in a human dopaminergic cell line [28,53]. Similarly, α-synuclein overexpression also causes microtubule destabilization and neuritic degeneration in cultured cells [54].

Misfolded and damaged polypeptides such as α-synuclein oligomers and aggregates in PD are mainly degraded by two major protein degradation systems—the autophagy-lysosomal pathway (ALP) and the ubiquitin-proteasome system (UPS) [55]. However, in the brains of patients with PD, lysosomal activity [56] and various lysosomal markers (LAMP1, heat-shock protein 73, and cathepsin D) were significantly decreased, especially in nigral neurons containing α-synuclein inclusions [57]. Consistently, dementia with Lewy body (DLB) brains also exhibits ALP depletion [58]. Additionally, α-synuclein aggregates inhibit macroautophagy by binding with essential ALP components, such as p62 and LC3 [58]. In cultured cells, chaperone-mediated autophagy was inhibited by mutated α-synuclein expression [59]. This inhibition results from α-synuclein binding with the specific receptor on the lysosome LAMP2, which might induce α-synuclein aggregation and injure the entire homeostasis of cells [60]. UPS dysfunction is also present in PD brains, such as decreased proteasome enzymatic activities and structural defects of proteasomes [61,62], and proteasome dysfunction is considered to be of primary or secondary consequence in a wide array of chronic neurodegenerative diseases [63,64,65,66,67]. As reported previously, both wild-type α-synuclein and mutant αSyn (A53T) expression resulted in a time-dependent and significant decrease in proteasome dysfunction in cellular models, such as SH-SY5Y cells [68] and PC12 cells [69]. A possible hypothesis about UPS inhibition by α-synuclein is the steric blocking of the proteasome machinery. Seventy percent of Lowy bodies, extracted from post-mortem DLB brains, are positive for 20S proteasome components [70]; more importantly, the 20S component was found in α-synuclein aggregates extracted from pre-formed α-synuclein fibrils-treated and α-synuclein-expressing HEK293 cells [71] and the 26S proteasome was isolated together with α-synuclein oligomers [72]. Additionally, aggregated proteins such as amyloid-β [70], tau [73], and α-synuclein [72] interact with and impair proteasome functions, and aggregated proteins such as α-synuclein and tau oligomers impair 20s proteasome function through allosteric impairment of the substrate gate in the 20S core particle and prevents the 19S regulatory particle from injecting substrates into the degradation chamber [73,74]. Altogether, these findings suggest that α-synuclein, especially α-synuclein oligomers, can block the UPS, thereby, damaging the clearance of other substrates and unbalancing proteostasis.

### 2.3. α-Synuclein and the Mitochondrial Surface

The inherent ability of α-synuclein to bind to lipids and membranes, especially membranes with negatively charged surfaces, increases the potential for interaction with the mitochondrial membrane [75]. Thus, the properties of α-synuclein in lipid binding were extensively investigated [76]. Several studies indicated that the ability for α-synuclein docking to the lipid surface is regulated by the deletion or insertion of charged amino groups in the first 25 residues of the N-terminal domain, adopting an α-helix conformation [77,78,79]. Thus, some mutations of α-synuclein, such as A30P, E46K, and A53T, in this domain might strongly affect this ability. Regarding the mitochondrial outer membrane (MOM) proteins, which α-synuclein are associated with, the in vitro pulldown experiments of extracts of the mitochondria with α-synuclein peptide composed of the last 40 residues of C-terminal retrieved TOM22, TOM40, voltage-dependent anion channels (VDAC)-1, 2, 3, and Samm50 as binding partners (Figure 2). However, the S129 phosphorylation distinctly inhibited α-synuclein association with TOM40 and Samm50 [80]. VDAC is a vital and recurrent binding partner of α-synuclein, and co-immunoprecipitation analysis revealed an association between the two proteins in the brainstem, striatum, and cerebral cortex in A53T human α-synuclein transgenic mice [81]. Moreover, recombinant α-synuclein monomers in vitro suggested marketable inhibition ability in the VDAC-1 channel activity in planar lipid bilayers [82].

Therefore, although α-synuclein is the able to bind membranes directly, especially the proteins that mirror the composition of mitochondria, and the protein–protein interactions with the MOM were consistently reported, the mechanism for the mitochondrial protein import machinery, particularly TOM20, remains unclear.

### 2.4. α-Synuclein in Mitochondrial Injury

The association between PD and mitochondria was reported in 1976; 1-Methyl-1,2,3,6-tetrahydropiridine (MPTP) was discovered as a mitochondrial complex I inhibitor, resulting in PD-like motor symptoms. Furthermore, complex I activity decreased by approximately 30% to 40% in the substantia nigra pars compacta (SNpc) of patients with PD. This further supports the vital role of mitochondria in PD development [83]. In contrast, experimental rodent models and patients with PD indicated strong evidence of damage to mitochondrial dynamics and increased reactive oxygen species (ROS) [84,85]. The main PD-connected genes are involved in mitochondrial homeostasis. Mutations in these genes are associated with mitochondrial dysfunction and might cause familial PD, as reported elsewhere [84]. New findings also described the role of α-synuclein in mitochondrial injury [86,87]. It is possible that there is a salient relationship between α-synuclein and mitochondria under physiological or pathological conditions.

Recently, the toxicity of α-synuclein in the mitochondria was extensively studied. Constitutive low α-synuclein levels in the mitochondria might play an essential role in maintaining normal functions of mitochondrial complexes. However, high levels of α-synuclein accumulation in the mitochondria contribute to the impairment of complex I function [12]. Furthermore, overexpression of wild-type or A53T mutant α-synuclein in SHSY-5Y cells or treating aggregated α-synuclein in an isolated mitochondrion, induces cytochrome c release, thereby increasing mitochondrial oxidative stress and causing mitochondrial dysfunction and fragmentation [11,87,88]. Although the mechanism through which α-synuclein translocates to the mitochondria remains unclear, the N-terminal domain of α-synuclein mimics mitochondrial-targeting sequence properties and contributes to the anchoring of α-synuclein to the mitochondrial membrane [88]. Additionally, α-synuclein suggested a high-affinity binding to the TOM20 pre-sequence receptor of the mitochondrial protein import machinery. When α-synuclein translocates to the mitochondrial membrane, it disrupts the translocase of the inner and MOM complex assembly process and impairs mitochondrial protein import from the cytosol [87,89,90,91]. Interestingly, α-synuclein also functions in the mitochondrial matrix, indicating that physiological α-synuclein accelerates ATP synthase activity by directly binding to its subunit, thereby ensuring mitochondrial health and proper ATP fueling for synaptic function [87]. Ludtmann et al. reported that α-synuclein monomers interact with and enhance ATP synthase under normal conditions [91]. Under pathological conditions, α-synuclein aggregation is stimulated and generates accumulated α-synuclein oligomers with increasing enrichment of β-sheet structures that are in close proximity to the inner mitochondrial membrane. They inhibit complex I function, thereby triggering oxidation events and ultimately leading to mega-channel opening and mitochondrial permeability transition events [91]. Therefore, α-synuclein itself does not interfere with the normal functions of mitochondria, and the toxicity of α-synuclein in the mitochondria is regulated by its oligomerization and aggregation events, which are associated with other pathological factors.

## 3. FABPs

### 3.1. Expression and Functions of FABPs in the Brain

In cells, fatty acids with a small number of carbon atoms are water-soluble; however, long-chain fatty acids with 12 or more carbon atoms are insoluble. Therefore, to move long-chain fatty acids inside the cells, an FABP that binds to and solubilizes long-chain fatty acids is required. The FABP family is a low-molecular-weight intracellular protein, with a molecular weight of 14–15 kDa, and up to 12 types of molecular species were identified in mammals [92]. They serve intracellular lipid chaperones and regulate the uptake and transportation of long-chain FAs [93,94], such as eicosanoids and bile acids. Their functions are thought to include intracellular uptake of ligands, transport to lipid metabolism pathways, and regulation of lipid signal transduction [95]. FABP is named after the first isolated tissue or cell, heart-type (H-), brain-type (B-), epidermis-type (E-), adipocyte-type (A-), liver-type (L-), intestinal-type (I-) FABP, etc. However, due to the diversity of tissue and cell expression, they were recently called L-FABP (FABP1), I-FABP (FABP2), and H-FABP (FABP3), in the order of isolation.

Among them, three FABPs, the heart-type (FABP3), brain-type (FABP7), and epidermal-type (FABP5), are expressed in the brain. FABP7 localizes to the neural stem cells of astrocytes and the hippocampal granule cell layer. This suggests high expression levels in the embryonic brain after birth; however, the expression levels decrease in the mature brain, astrocytes, and OPCs [96,97]. This expression is localized in astrocytes and the hippocampal neural stem cells [98]. Analysis using FABP7 gene knock-out mice revealed that FABP7 is involved in n-3 long-chain unsaturated fatty acid uptake into astrocytes.

Interestingly, no abnormalities were observed in spatial memory and learning in FABP7-deficient mice, but aversive memory, fear, and anxiety were enhanced [96]. On the other hand, FABP5 is most widely expressed in the epidermis, liver, and adipocytes and is similar to FABP7. It is highly expressed in the nerve and glial cells in embryonic brains but is decreased in mature brains [97,99,100]. The expression of mRNA peaks from neurogenesis (E17) to the process of nerve differentiation (E19), and its expression decreases during the synaptogenesis period (P5-P10), suggesting that it is involved in the migration and differentiation of nerve cells [101]. On the other hand, FABP3 is not expressed in an embryonic brain but is expressed in nerve cells in a mature brain [99]. FABP3 was first isolated from the heart, although it is widely expressed in the tissues and cells in the body, it was reported to be expressed in the brain, skeletal muscle, mammary gland, and ovary, in addition to the heart. In contrast to FABP7, FABP3 has a high affinity for n-6 unsaturated fatty acids, such as arachidonic acid [102]. It was also shown to bind to epoxyeicosatrienoic acid, an arachidonic acid metabolite, and to prolong its half-life [103]. Furthermore, it was reported that FABP3 expression is increased in the serum and cerebrospinal fluid (CSF) of patients with neurodegenerative disorders, including PD, AD, and other neurological disorders [104,105], suggesting that it might not only be related to normal brain function but also to neurological disease.

### 3.2. Functions of FABPs in Neurodegenerations

#### 3.2.1. FABP3 in the Dopamine Nervous System

The dopamine nervous system is deeply involved in the development of severe psychiatric disorders, such as schizophrenia and attention deficit hyperactivity disorder [106,107]. Most antipsychotics have a D2 receptor blocking effect, and blocking the striatal D2 receptor is effective in improving positive symptoms in schizophrenia [108]. The D2 receptor has two subtypes, D2L and D2S, which depend on the presence or absence of the 29 amino acid residues in the third loop in the cell [109]. D2L receptors are mainly expressed at the posterior synapses of the striatum and the marginal system, and D2S receptors are present as auto-receptors in the substantia nigra and dopaminergic nerve endings (pre-synapses) in the ventral tegmental area. Previously, we investigated the intracellular expression and signal transduction of the NG108-15 cells, which specifically express the D2L and D2S receptors [110,111,112]. Different localizations of D2L and D2S receptors were found, in which most D2S receptors are expressed on the cell membrane but the D2L receptor is strongly expressed around the inner core [112]. Using FABP3 knockout mice, it was found that FABP3 binds to the D2L receptor and regulates D2L receptor function in dopamine-related locomotor activity and motor coordination [113].

#### 3.2.2. FABP3 in α-Synuclein Oligomerization and Migration

FABP3 binds to the D2L receptor [113,114,115]. It suggests that FAPB3 plays potential regulatory functions in the dopamine nervous system related to the process of PD. Recently, the positive function of FABP3 in α-synuclein accumulation was demonstrated. In the MPTP-induced mouse model of PD, FABP3 is implicated in MPTP-induced neuronal toxicity and α-synuclein accumulation [13]. First, it was determined that FABP3 knockout mice were more resistant to MPTP toxicity and suggested weaker dopamine neuronal loss and motor deficits in a mouse PD model, as compared to wild-type mice. Then, enhanced α-synuclein oligomerization in response to elevated FABP3 levels was observed; however, in FABP3 knockout mice, α-synuclein oligomerization remained markedly decreased. Moreover, FABP3-mediated AA incorporation following neurotoxin exposure induces α-synuclein oligomerization. Importantly, α-synuclein binds to FABP3 directly via its C-terminal region and enhances the formation of toxic oligomers (Figure 3) [116]. Additionally, inhibition of FABP3 by FABP3 ligands also significantly attenuated α-synuclein oligomerization [117], dopamine neuronal loss, and behavioral impairments [118].

In contrast, FABP3 also plays critical roles in extracellular α-synuclein monomer uptake and is critical for MPTP-induced neuronal retraction, mitochondrial function, and oxidative stress [14]. To elucidate the mechanism through which FABP3 mediates α-synuclein uptake, mesencephalic neurons derived from dopamine D2L knockout mice, FABP3 knockout mice, and wild-type C57BL6 mice were employed and the ability of fluorescence-conjugated α-synuclein monomers and fibrils uptake was measured in a previous study [119]. The physiological significance of D2L in α-synuclein uptake in dopaminergic neurons in D2L knockout mice and wild-type C57BL6 mice was investigated and the uptake ability of ATTO-55O-labeled α-synuclein monomers in D2L knockout neurons was observed. This demonstrated that the D2L receptor is vital for α-synuclein uptake into dopaminergic neurons. Since dopamine D2 receptors are abundantly localized in membrane rafts [120] and internalized via the caveolae-mediated endocytic pathway [121], the cells were further treated with dynasore or interfered with caveolin-1 expression by caveolin-1 shRNA, thereby inhibiting the caveolar function. Abolished uptake of α-synuclein monomers after caveolar inhibition was found. As with α-synuclein monomers, the uptake of α-synuclein fibrils by FABP3 knockout mice and D2L knockout mice was also identified, and both FABP3 and D2L receptors were found to be vital for the uptake of α-synuclein fibrils. The results indicated that dopamine D2L with a caveola structure, coupled with FABP3, plays an essential role in both α-synuclein monomer and fibril uptake by dopaminergic neurons, suggesting a novel pathogenic mechanism of synucleinopathies, including PD [119].

#### 3.2.3. FABP5 in Cognitive Deficits

Docosahexaenoic acid (DHA) is a vital n-3 polyunsaturated fatty acid in the brain, suggesting potent effects on cognition function, by regulating synaptic transmission and long-term potentiation in the hippocampus [122,123]. Due to the limited ability of the brain to synthesize its own DHA, DHA transports across the blood–brain barrier (BBB) to reach the brain [124,125]. In support of this pathway, FABP5 was reported to be a critical protein involved in the uptake of exogenous DHA [126]. Furthermore, a recent study also indicated that FABP5 is localized in the BBB and is essential for maintaining brain endothelial cell uptake of DHA, and FABP5 knockout mice suggested obvious cognitive deficits associated with decreased DHA levels in the brain [127]. On the other hand, another study indicated cognitive impairments in learning and memory in FABP5 knockout mice, which are related to diminished peroxisome proliferator-activated receptor (PPAR) β/δ activation by arachidonic acid [128].

On the other hand, in experimental autoimmune encephalomyelitis (EAE) in a mouse model of MS, the deficiency of FABP5 indicated a protective effect from the development of EAE [129,130]. Dendritic cells with FABP5 deficiency are defective in the production of proinflammatory cytokines and Th1 and Th17 responses [130]. In addition, FABP5 deficiency in CD4+ T cells exhibited increased expression levels of PPARγ, which inhibited Th17 differentiation and enhanced regulatory T cell development [129]. Importantly, inhibition of FABP5 by inhibitors can regulate the functions of T cells and antigen-presenting cells, thereby ameliorating the clinical symptoms of EAE by inhibiting pathogenic processes and lymphocyte migration [131]. Importantly, in our recent study, we also found a tight connection between FABP5 and α-synuclein [132]. In Neuro-2A cells treated with rotenone, α-synuclein forms high molecular weight α-synuclein oligomers, together with FABP5 in SDS-soluble fractions and triggers significant loss of cell viability. This might indicate a potential role of FABP5 in dopaminergic neuron loss in synucleinopathies.

#### 3.2.4. FABP5 in Mitochondrial Injury

Mitochondria are critical organelles for fatty acid metabolism, and FABPs are chaperone proteins that bind hydrophobic ligands, thereby coordinating lipid uptake and intracellular trafficking [133]. Several previous studies demonstrated the critical roles of FABP5 in maintaining normal mitochondrial integrity and function, via regulation of lipid metabolism in Treg cells [134] and memory T cells [135]. However, in our recent study, we demonstrated a potential mechanism through which FABP5 might be connected with mitochondrial injury in neurodegeneration. In psychosine-treated (a critical biochemical pathogenetic mechanism of the loss of oligodendrocytes and myelin in KD) oligodendrocytes, we found abnormal accumulation of FABP5 in the mitochondria that formed macropores in the MOM associated with VDAC-1 and BCL-2-like protein 4 (BAX) [18]. Furthermore, along with macropore formation, inflammatory substances in mitochondria, such as mitochondrial DNA (mtDNA), cytochrome C release to the cytosol, and trigger activation of cysteine proteases, thereby, causing cell death. These data indicate a potential role of FABP5 in inducing oligodendrocyte loss in KD (Figure 4).

In our recent study, we also found that FABP5 normally localized in the mitochondria of Neuro-2A cells. Interestingly, FABP5 co-localized with α-synuclein and formed hetero-oligomers and aggregates in α-synuclein-overexpressing Neuro-2A cells treated with rotenone [132]. Furthermore, we also observed that rotenone treatment led to α-synuclein aggregate localization in the mitochondria and decreased mitochondrial membrane potential, resulting in cell death [132]. This finding indicated that FABP5 is involved in mitochondrial injury associated with α-synuclein in synucleinopathies.

#### 3.2.5. FABP7 in Neurodegenerations

Several studies indicated that FABP7 is also associated with pathological processes in the CNS. In young adult monkeys, the expression levels of FABP7 increased in different hippocampal sub-regions in response to cerebral ischemia [136]. The transient, complete, and whole-brain ischemia model of adult monkeys, which was induced by clamping the innominate and left subclavian arteries for 20 min, suggested elevated expression levels of FABP7 in the hippocampus and were co-localized to proliferating cells following ischemia. [137]. In addition, FABP7 expression is strongly upregulated and co-localized with GFAP expression in response to contusive spinal cord injury in mice [138]. On the other hand, in MOG-induced EAE mice, FABP7 and GFAP double-positive cells accumulated within the spinal cord [139], and FAB7 knockout mice exhibited attenuated clinical scores in the late phase of EAE [140].

## 4. Novel Therapeutic Target to Mitochondrial Dysfunction via FABP Inhibition

Although α-synuclein monomers normally bind to the mitochondrial membrane, suggest tight interactions with ATP synthase, and improve ATP synthesis efficiency [91], some other reports also indicated the toxic effect of α-synuclein monomers on mitochondria. In SHSY-5Y cells, both wild-type and A53T mutant α-synuclein overexpression induced significant cytochrome c release and mitochondrial calcium and nitric oxide increase [11]. However, the A53T mutant α-synuclein monomers showed a higher tendency to localize to the mitochondrial membrane and higher toxicity than the wild-type [141]. In contrast, α-synuclein translocated to the mitochondria in response to cellular stress and directly upon intracellular acidification [142], dysregulation of ion transport [143], hydrolysis of high-energy nucleotides [144], and the PD toxin MPP^+^ impaired cellular energy metabolism and resulted in a decrease in pH [145]. The mutant α-synuclein-like A53T is more prone to aggregation, attenuates oxidative stress [146,147], and interferes with iron homeostasis [148,149]. Inhibition of α-synuclein aggregation is important for maintaining iron homeostasis and normal ROS and pH levels, thereby protecting the mitochondria from α-synuclein toxicity in neurodegeneration. The inhibition of FABP3 by FABP3 ligands significantly disrupted α-synuclein oligomerization and aggregation in both cultured Neuro-2A cells [117] and mouse models [118]. Furthermore, the inhibitor of FABP5 by FABP5 inhibitor also inhibited FABP5-dependent α-synuclein oligomerization towards mitochondrial localization and toxicity [132].

In contrast, FABP5 is vital for macropore formation to interact with VDAC-1 oligomers and BAX, under psychosine stress [18]. Macropores are usually involved in the abnormal release of mtDNA and cytochrome c, which are connected with the activation of the cGAS-STING pathway and mitochondrial damage [77]. We recently reported a novel FABP5 inhibitor, which suggested a high affinity for FABP5 and blocked FABP5-dependent VDAC-1 oligomerization, thus, inhibiting macropore formation. Furthermore, the FABP5 inhibitor also decreased cytochrome c release from mitochondria, attenuated the generation of inflammatory cytokine-like IL-1β in the cytosol, and rescued mitochondrial dysfunction in oligodendrocytes.

## 5. Conclusions

The etiology of NDD is likely multifactorial. Many factors might be involved in this, including mitochondrial dysfunction, clearance system injury (proteasome and lysosome dysfunction), endoplasmic reticulum stress, and interfered synapse transmission. Here, we reviewed the mitochondrial injury associated with α-synuclein and FABPs in NDDs, and provided a novel therapeutic target for mitochondrial dysfunction.

α-Synuclein is capable of binding to the MOM, directly by the N-terminal [77,78,79] and interaction with MOM proteins, such as TOM20 and VDAC-1. However, α-synuclein binds to TOM20, which is involved in impairing the TOM20/TOM22 assembly, leading to injured respiration and ROS production [89]. Consistent with TOM20, VDAC-1/α-synuclein association also inhibited VDAC-1 activity in vitro. In contrast, a recent study also reported a positive effect of α-synuclein monomers on ATP production, and only the oligomeric forms are toxic to mitochondria [91]. Recently, several studies indicated the role of FABP3 and FABP5 in α-synuclein oligomerization and migration, as well as triggering the loss of mitochondrial function in neurons [13,14,118,135]. Importantly, FABP5 is a vital protein that interacts with VDAC-1 and forms mitochondrial macropores under psychosine treatment [18]. Therefore, inhibition of FABP3 and FABP5 is novel and critical for the treatment of NDDs, which is attributed to mitochondrial injury.

In the present review, we summarized the molecular mechanism of α-synuclein and its aggregation in neuronal toxicity and mitochondrial injury. We also reviewed a new protein family, FABPs, as risk factors for inducing α-synuclein aggregation and mitochondrial injury. These insights might lead to more focused efforts to develop therapeutics and strategies to prevent the onset of NDDs targeting mitochondrial injury.

## Figures and Tables

**Figure 1 biomedicines-09-00560-f001:**
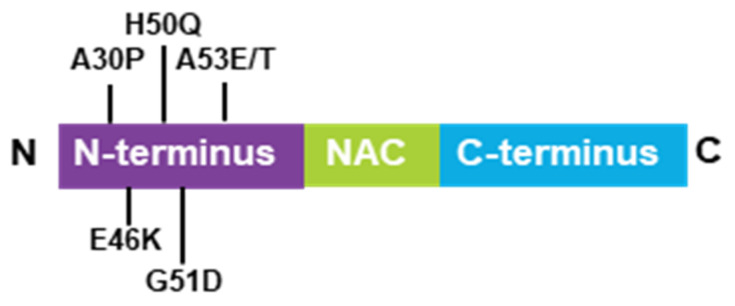
Schematic model of the α-synuclein structure, containing the N-terminal domain, NAC, and the C-terminal domain. Six synucleinopathy-related point mutaions described so far.

**Figure 2 biomedicines-09-00560-f002:**
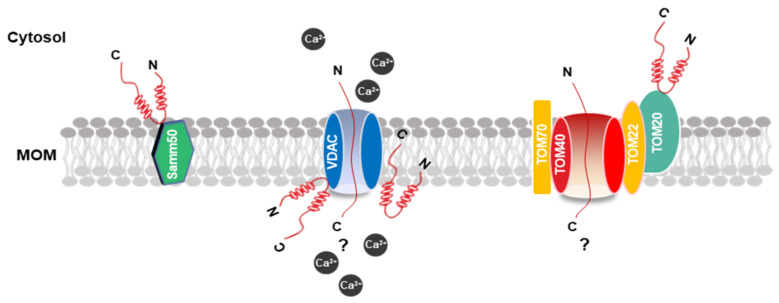
Schematic model of the MOM and α-synuclein association. The picture shows α-synuclein binding patterns at the MOM. Proteins that interact with α-synuclein are highlighted in colors. The possible mechanism of α-synuclein transport into the mitochondria through VDAC and TOM20. α-synuclein is shown in red. MOM, mitochondrial outer membrane; VDAC, voltage-dependent anion channels.

**Figure 3 biomedicines-09-00560-f003:**
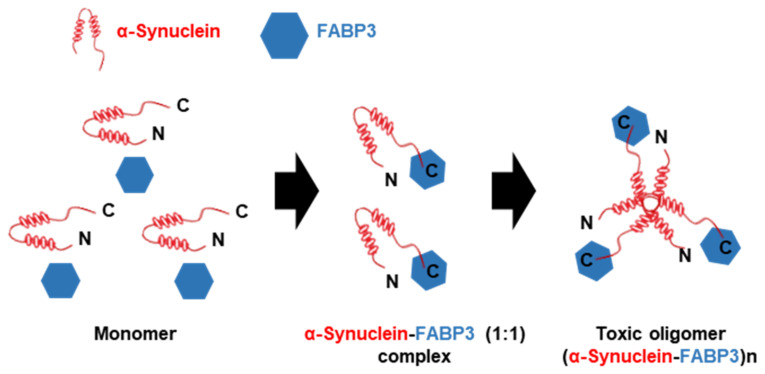
Schematic model of toxic α-synuclein oligomerization with FABP3. α-synuclein initially exhibits as a monomeric form in solution. However, in the presence of FABP3, α-synuclein binds to FABP3 via its C-terminal region and forms a soluble α-synuclein-FABP3 (1:1) complex. In addtion, the α-synuclein-FABP3 complex changes over time to oligomeric forms, (α-synuclein-FABP3)n that displays cytotoxicity (modified from [116]).

**Figure 4 biomedicines-09-00560-f004:**
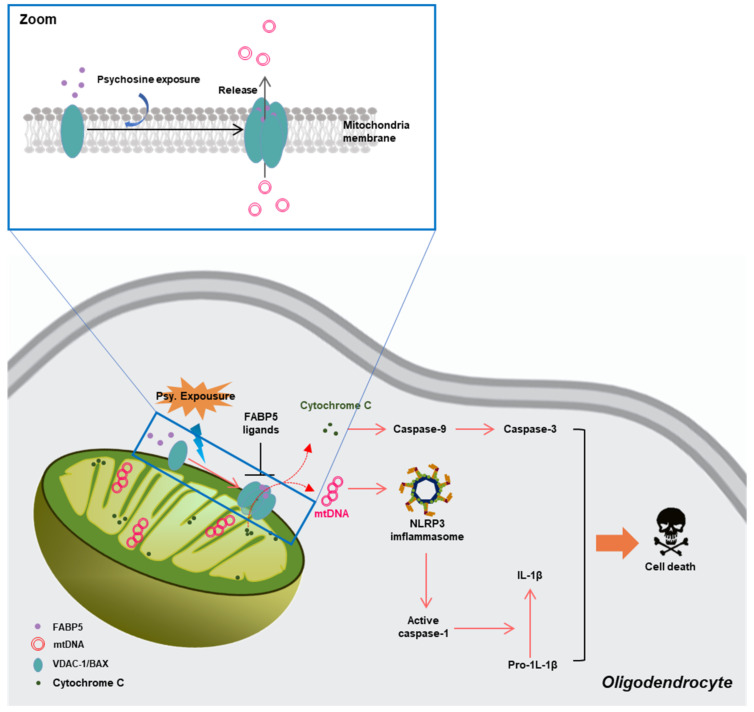
Schematic representation of the pathways through which FABP5 facilitates mitochondrial macropore formation and induces oligodendrocyte apoptosis [18]. FABP—fatty acid-binding proteins.

## Data Availability

Not applicable.

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
