# Peer review of "Impact of Fatty Acid-Binding Proteins in α-Synuclein-Induced Mitochondrial Injury in Synucleinopathy"

_biomedicines, 2021, doi:10.3390/biomedicines9050560_

Round 1

Reviewer 1 Report

The authors present a well put together summary of the a-Syn and its role in mitochorndrial injury and outline the impact of this on neurodegenerative disorders. The Introduction section and the Synulcien sections are well written with appropriate background and references that make for a logical read. However, these are some of my comments:

1) A figure showing a linear structure of a-Syn and annotating all the key mutations, and pointing out regions of interaction with FABPs would be very useful to include in this section.

2) The section about FABPs overall is also well supported by recent studies but lacks a smooth logical flow. There are quite a few instances where the choice of words throws off the reader, for example: Line 219, the sentence starts with "Furthermore, FABP3...." when the previous lines are about FABP7 and nothing has been said about FABP3. Another example is line 248 where the sentence starts with "Since FABP3..." this is a grammatically incorrect sentence. So I suggest the authors to revise these errors for a more logical reading.

3) A minor comment: Section 3.2.3 talking about Mitochondrial injury. This section does not really add much to the rest of the article and does not strictly need to be included.

4) The figure 2 needs to be of a better resolution. The image sizes are too small to really be useful.

Overall, this is a very informative review article and is of interest. However, the authors need to implement these minor changes to make the article more easily comprehensible to the reader.

Author Response

# Reviewer 1:

The authors present a well put together summary of the a-Syn and its role in mitochondrial injury and outline the impact of this on neurodegenerative disorders. The Introduction section and the Synulcien sections are well written with appropriate background and references that make for a logical read. However, these are some of my comments:

1) A figure showing a linear structure of a-Syn and annotating all the key mutations, and pointing out regions of interaction with FABPs would be very useful to include in this section.

Answer: We add structure of a-Syn and mutations in Figure 1 (Page 2). The interaction between FABP3 and C-terminal regions of a-Syn is provided in Figure 3 (Page 5) and we described this part in line 258-260.

2) The section about FABPs overall is also well supported by recent studies but lacks a smooth logical flow. There are quite a few instances where the choice of words throws off the reader, for example: Line 219, the sentence starts with "Furthermore, FABP3...." when the previous lines are about FABP7 and nothing has been said about FABP3. Another example is line 248 where the sentence starts with "Since FABP3..." this is a grammatically incorrect sentence. So I suggest the authors to revise these errors for a more logical reading.

Answer: According to the comment, we corrected it (line 219; line 248-250).

3) A minor comment: Section 3.2.3 talking about Mitochondrial injury. This section does not really add much to the rest of the article and does not strictly need to be included.

Answer: According to the comment, we removed this section about mitochondrial injury and FABP3.

4) The figure 2 needs to be of a better resolution. The image sizes are too small to really be useful.

Answer: According to the comment, we corrected it (Figure 4) (page 8)

Overall, this is a very informative review article and is of interest. However, the authors need to implement these minor changes to make the article more easily comprehensible to the reader.

Reviewer 2 Report

Well written review conecerning  molecular mechanism of α-synuclein and
its aggregation in neuronal toxicity and mitochondrial injury. Additionally describing FABPs, as risk factors for inducing α-synuclein aggregation and mitochondrial injury. I really appreciate comprehensive approach of authors. 

My only comment is to improve quality of Figures.  Too small fonts and illegible in total, because of too small format. Needs improvent. 

Author Response

# Reviewer 2:

Well written review concerning molecular mechanism of α-synuclein and
its aggregation in neuronal toxicity and mitochondrial injury. Additionally, describing FABPs, as risk factors for inducing α-synuclein aggregation and mitochondrial injury. I really appreciate comprehensive approach of authors. 

My only comment is to improve quality of Figures.  Too small fonts and illegible in total, because of too small format. Needs improvent. 

Answer: According to the comment, we corrected all the figures of the manuscript.